# In-Depth Comparison of Regularization Methods For Long-Tailed Learning in Trajectory Prediction

## Abstract

Autonomous robots have the biggest potential for risk because they operate in open-ended environments where humans interact in complex, diverse ways. To operate, such systems must predict this behaviour, especially if it's part of the unexpected and potentially dangerous long tail of the dataset. Previous works on long-tailed trajectory prediction use models which do not predict a distribution of trajectories with likelihoods associated with each prediction. Furthermore, they report metrics which are biased by the ground-truth. Therefore, we aim to examine regularization methods for long-tailed trajectory prediction by comparing them on the KDE metric, which is designed to compare distributions of trajectories. Moreover, we are the first to report the performance of these methods on both the pedestrian and vehicle classes of the NuScenes dataset.

## 1 Introduction

A major challenge of predicting future trajectories in open-ended environments (i.e. environments where an agent's future goal or path can take on an unbounded number of possibilities) is that the behaviors encountered resemble a long tailed distribution. There are many examples of easily predictable behaviors like standing still or walking at a constant speed, and few examples of complicated behaviors like turning to go into a store. Although the issue of long-tailed learning is well studied in classification problems, improving the long tail in regression is much more complicated, especially in tasks like trajectory prediction (Thuremella & Kunze, 2023a).

In this work, we focus on long-tailed learning methods within trajectory prediction, and compare two regularization methods developed for long-tailed learning: that of Makansi et al. (2021) and Kozerawski et al. (2022). To our knowledge, these are the only two regularization methods developed for long-tailed learning within trajectory prediction. Due to the fact that both of these methods have only been applied to non-probabilistic trajectory prediction approaches and evaluated on minADE/minFDE metrics (metrics which evaluate only the best of many predicted trajectories against the ground truth), we apply these methods to the probabilistic trajectory prediction approach Trajectron++ Salzmann et al. (2021), and evaluate them on both pedestrians and vehicles within the NuScenes dataset (which was not previously done). Furthermore, we discuss the efficacy of the different strategies applied by these two methods using our results.

Our contributions include: 1) re-evaluating regularization methods for long-tailed trajectory prediction on both pedestrians and vehicles within NuScenes, on more traditional metrics such as most likely FDE, and KDE, as in Salzmann et al. (2021), and 2) comparing the efficacy of the two methods described by employing both quantitative and qualitative comparisons.

## 2 Background

### 2.1 Trajectory Prediction

Trajectory prediction is a regression task, where a series of coordinates that correspond to an agent's future location are predicted using their past location, sometimes in combination with other features like maps (Salzmann et al., 2021). Agent history is typically represented as a set of past location

coordinates (e.g. Sadeghian et al., 2018), and maps are usually represented as rasterized images with semantic layers (Caesar et al., 2020). Multimodality in trajectory prediction is a large area of interest (e.g. Dong et al., 2021; Kosaraju et al., 2019; Gu et al., 2022) and has been widely studied using both probabilistic methods like (conditional) variational auto encoders (VAEs or CVAEs) (e.g., Zhou et al., 2021; Xu et al., 2022) and deep neural net training techniques (e.g., Makansi et al., 2019).

## 2.2 Long-Tailed Learning

Most naturally sampled datasets that contain many examples of a few common cases and few examples of many uncommon cases are long-tailed. The uncommon examples in the long tail are harder to predict, as they are rare and dispersed among the many majority cases. Many classification surveys have covered the plethora of long-tailed learning techniques within the various classification problems like image recognition (e.g., Zhang et al., 2021), action recognition (e.g., Özyer et al., 2021; Vrigkas et al., 2015; Yadav et al., 2021), and action prediction (e.g., Rasouli et al., 2020b; Xu et al., 2020; Rasouli et al., 2020a; Zaech et al., 2020). However, dealing with imbalanced datasets in regression is more complicated, especially in multidimensional regression tasks like trajectory prediction, because defining a metric by which to determine whether an example falls into the long tail is non-trivial (Thuremella & Kunze, 2023a)

## 2.3 Long-Tailed Learning in Trajectory Prediction

Makansi et al. (2021), Kozerawski et al. (2022), and Wang et al. (2023) directly address long-tailed learning in trajectory prediction, with Wang et al. (2023) using a mixture of experts and Makansi et al. (2021) and Kozerawski et al. (2022) using regularization techniques, while Li et al. (2021) simply show that injecting logic rules by adding cross-walks, traffic lights, and left/right turn only lanes into the map and making them hard rules instead of suggestions to be used as input, reduces the long tail of the error distribution, as shown in Figure 3 of Li et al. (2021). Although Anderson et al. (2019) don't directly address dataset imbalance, they develop a data augmentation method that could be used to upsample uncommon trajectories by generating trajectories from dataset statistics and adding random transformations to increase the variety and number of trajectories.

## 3 Method

## 3.1 Dataset

To train and evaluate our model, we use the NuScenes dataset Caesar et al. (2020), which consists of 1000 scenes with 5.5 hours of footage labeled at 2Hz. It has 17,081 labeled tracks taken from a moving vehicle in 4 neighborhoods within Boston and Singapore (boston-seaport, singapore-onenorth, singapore-queenstown, singapore-hollandvillage), and includes HD semantic maps with 11 annotated layers, including pedestrian crossings, walkways, stop lines, traffic lights, road dividers, lane dividers, and driveable areas Caesar et al. (2020).

## 3.2 Models

The baseline model we use to compare long-tailed learning methods is Trajectron++ Salzmann et al. (2021), as it produces a multi-model distribution of future trajectories and their likelihoods, which is useful in planning applications. Furthermore, this work is referenced in many other papers as a point of comparison, since it significantly advanced the state of the art.

Trajectron++ implements multi-modality by forming a probabilistic idea (supported by a Gaussian Mixture Model) of the distribution of the future trajectory space and samples this distribution in order to obtain any number of future predictions. To perform trajectory prediction, Salzmann et al. (2021) concatenates the map encoding, history encoding, and social influence encoding into a single learned feature representation, and then uses this representation as the input to a CVAE model in order to learn a latent space embedding, which is then used to predict future positions iteratively using a GRU, as shown in Figure 1. The goal of the CVAE is to explicitly handle multimodality and allow the latent space embedding to learn high level latent behavior Salzmann et al. (2021).

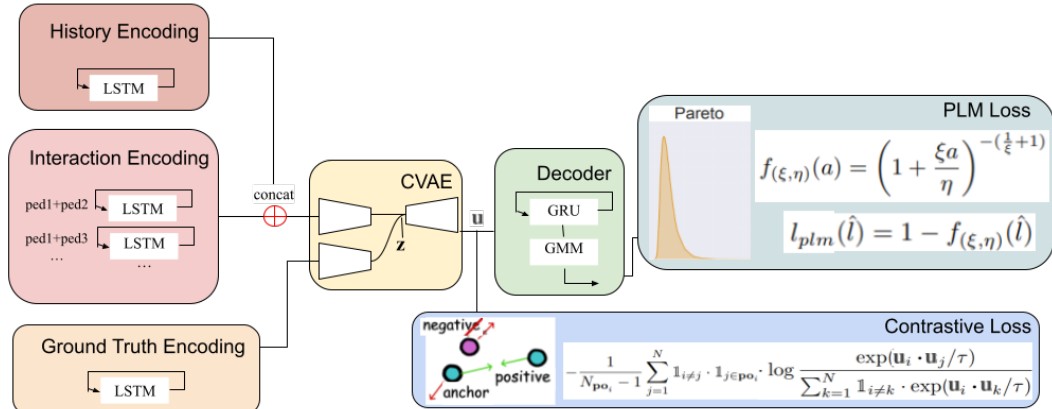

Figure 1: Architecture of Baseline Model with contrastive (Makansi et al., 2021) and PLM re-weighting (Kozerawski et al., 2022) long-tailed learning techniques. Contrastive loss pushes the embeddings of nodes in the same class (i.e. similar 'difficulty' level) together, and those of different classes apart, as shown (where $\tau$ is a pre-defined hyperparameter and $\mathbf{po}_i$ is the positive set of anchor $i$, i.e. the set of samples $j$ in the batch which has a difficulty score $s_j$ satisfying $|s_i - s_j| < \theta_p$, where $\theta_p$ is a hyper-parameter defining the positivity threshold). PLM re-weighting loss takes the initially calculated per-example loss, $\hat{l}$, and uses the assumption that the long tail is shaped like a pareto curve to transform it according to the equation shown (where $\xi$ and $\eta$ are pre-defined hyperparameters). The diagrams of the baseline model architecture are based on Salzmann et al. (2021), while pictures and equations of the contrastive and PLM losses are taken from Makansi et al. (2021) and Kozerawski et al. (2022) respectively. The diagram for contrastive loss is from Dee.

The CVAE accomplishes this by using the ground truth future trajectory within the model to learn the latent space embedding. As shown in Figure 1, one branch of the model estimates the latent space embedding using the concatenated feature representation while a second branch estimates the latent space embedding using the ground truth future trajectory and the feature representation. Then, the CVAE loss minimizes the difference between the two latent space embedding estimates using Kullback-Leibler divergence loss. During inference, however, only the branch which uses just the feature representation is employed to create the latent space embedding which is then fed to a GRU and a Gaussian Mixture Model to decode the embedding into future trajectory positions.

In contrast, past long-tailed trajectory prediction methods ( Makansi et al. (2021) and Kozerawski et al. (2022)) have used the Trajectron++EWTA model Makansi et al. (2021) as a baseline since its minADE/minFDE metrics show better performance than Trajectron++ Salzmann et al. (2021). However, this comes at a cost: the EWTA (Evolving Winner-Takes-All) loss always predicts $N$ (in this case, 20) future trajectories without any associated likelihoods, and specifically optimizes for the 'Best-of-20' metric by 'evolving' the training scheme such that in the beginning, loss is averaged across all 20 trajectories, but by the end of the training, loss is only optimized for the single trajectory that is closest to the ground truth Makansi et al. (2019).

To train our baseline model, we maintain the same training methodology and parameters as Salzmann et al. (2021), and predict a distribution of trajectories that can be sampled with associated likelihoods. We train our model to predict 3s into the future using a history of 3s, and evaluate after 12 epochs, with a batch size of 256.

### 3.2.1 CONTRASTIVE LOSS

To improve long-tail performance, Makansi et al. (2021) use contrastive loss on implicit classes of trajectories to force the model to learn the characteristics of rare trajectories separately from common trajectories. This loss forces the feature embeddings of the rare trajectories to be pushed apart from the feature embeddings of common trajectories, in the feature space (Makansi et al., 2021), as shown in the contrastive loss diagram in Figure 1. Therefore, feature embeddings of rare

trajectories are less likely to be lost within the manifold of common trajectories, and assumed to be outliers. In Makansi et al. (2021), classes are defined by how easy it is to predict the future trajectory through a physics-based Kalman filter: rare and important trajectories are assumed to be the ones which are difficult to predict using simple kinematics.

We implement the contrastive loss proposed by Makansi et al. (2021) by taking the feature embedding from the output of the CVAE (before the decoder), and using it as the feature space on which to separate common examples from uncommon examples, as shown in Figure 1. All other parameters of the contrastive loss were taken from the default values in Makansi et al. (2021). A diagram of how this this loss regularizer is incorporated into the model is shown in Figure 1.

### 3.2.2 PLM Loss

Kozerawski et al. (2022), on the other hand, compare two novel loss terms that up-weight rare, high error examples: a regularization term which improves performance slightly in average and rare cases, and a kurtosis term which significantly improves only the worst error. The regularization term includes hyperparameters that assume a fixed shape for the error distribution (i.e. a pareto shape, as shown in Figure 1), while the kurtosis term uses batch statistics to estimate the error distribution.

We use the best method proposed by Kozerawski et al. (2022), the regularization term, by adding the PLM regularization function from Kozerawski et al. (2022) to the individual loss of each example. We use the same parameters as the default values in Kozerawski et al. (2022). The equation for how this this regularizer is incorporated into the loss is shown in Figure 1.

In addition to using the default parameters of Kozerawski et al. (2022) and Makansi et al. (2021) for the PLM loss and Contrastive loss, we also perform an ablation study to see how applying more or less regularization might affect the model. The results of this ablation study are in the Appendix.

## 4 Results

### 4.1 Metrics

Though the model was only trained to predict 3s into the future, we evaluate on predictions that are 3 and 4s into the future, as done in Salzmann et al. (2021) to demonstrate ability to generalize to more long-term prediction timeframes.

We follow most methods using NuScenes (e.g. Salzmann et al. (2021); Greer et al. (2021); Ghoul et al. (2022)) and use the final distance error (FDE) of the most likely predicted trajectory as our main evaluation metric. To bolster our results, we also evaluate our models on the KDE-NLL metric used in Salzmann et al. (2021) to show that performance of the entire distribution of predicted trajectories is improved, and not just that of the most likely final prediction. KDE NLL is the mean negative log-likelihood of the ground truth trajectory using the probability density function of a distribution found by fitting a kernel density estimate on trajectory samples Vishnu et al. (2023). Therefore, it takes into account the full trajectories of the multi-modal distribution of predictions.

While the FDE metric provides a tangible way to visualize the error (since it uses the physical units of meters), the KDE metric allows for better comparison between different methods because it takes into account not only the whole trajectory, but also the distribution of all possible futures that were predicted and their respective likelihoods. Therefore, we report both the FDE most likely metric, in order to facilitate visualization of error, and the KDE metric, in order to facilitate comparison.

### 4.1.1 Long-Tailed Metrics

To evaluate improvement in the long tail, we must also evaluate the performance of only the long tail of the dataset. However, the two methods presented above define and evaluate on two different sets of long-tailed metrics. While Makansi et al. (2021) uses a 'difficulty scoring' (based on how easy it is for a Kalman filter to predict the future trajectory) to get the ADE/FDE of the most 'difficult' 1, 2, and 3 percent of examples, Kozerawski et al. (2022) calculate the 95th, 98th, and 99th percentile of the distribution of errors to measure long-tail performance. This percentile is equivalent to measuring the CVaR (probability of predictions below a certain error), and used as a measure of risk in prediction works like Ren (2022) and Nishimura et al. (2023).

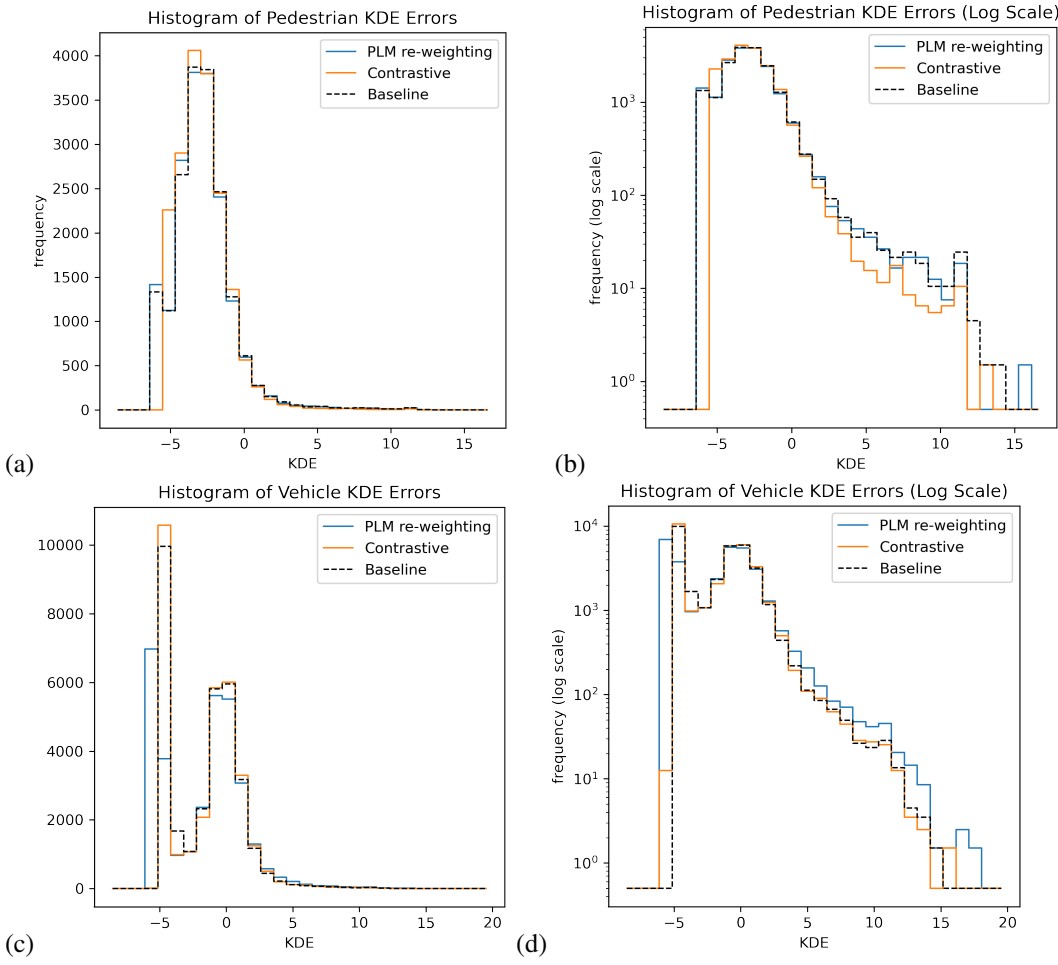

Figure 2: Histograms of pedestrian and vehicle KDE errors on NuScenes test set for each model, to facilitate comparison of long-tailed performance characteristics between models. (a) and (c) show the outlines of the frequency histogram (i.e. number of examples in the test set whose predicted trajectory distribution falls into the corresponding range of KDE error) for pedestrians (a) and vehicles (c), while the (b) and (d) show the same histograms on a log-scale to better highlight the differences between model performances within the long tail. When plotting the histograms on a log scale, a constant of 0.5 was added to the frequency count of each bin to prevent irregularities in the graph.

Although their definitions of long-tailed metrics are different, the metrics defined by Makansi et al. (2021) and Kozerawski et al. (2022) are equivalent, as shown by Thuremella & Kunze (2023b). Therefore, we follow the long-tail metrics defined by Kozerawski et al. (2022) because they are 1) simpler to visualize (as the 'difficulty' measure defined by Makansi et al. (2021) adds an extra layer of complexity), and 2) supported as a measure of risk and long-tailed performance by other works like Nishimura et al. (2023).

## 4.2 QUANTITATIVE EVALUATION

### 4.2.1 PEDESTRIANS

Since the metrics reported by Makansi et al. (2021) and Kozerawski et al. (2022) use a non-intuitive, difficult to visualize definition of FDE, i.e. minFDE, which calculates the error not of the most likely prediction, but of the best prediction out of 20 predicted trajectories, we report the most likely FDE, the final distance error of the single most likely prediction, as determined by the model. The most likely FDE performances of the baseline model (Salzmann et al., 2021), contrastive model

Table 1: Pedestrian FDE of most likely trajectories on NuScenes, predicting 3s and 4s into the future, where the columns are the average performance across the test set, and the 95th, 98th, and 99th percentile error across the test set.

| Model | Pedestrian FDE @3s | | | | Pedestrian FDE @4s | | | |
|---|---|---|---|---|---|---|---|---|
| | avg | 95th | 98th | 99th | avg | 95th | 98th | 99th |
| Baseline | 0.37 | 1.14 | 1.56 | **1.85** | 0.62 | 1.92 | 2.57 | 3.03 |
| Contrastive | **0.36** | **1.12** | **1.53** | 1.86 | 0.60 | **1.87** | **2.55** | **3.01** |
| PLM Re-Weighting | **0.36** | 1.15 | 1.54 | 1.87 | **0.59** | 1.91 | 2.56 | **3.01** |

Table 2: Pedestrian KDE on NuScenes, predicting 3s and 4s into the future, where columns are the average performance across the test set, and the 95th, 98th, and 99th percentile error across the test set

| Model | Pedestrian KDE @3s | | | | Pedestrian KDE @4s | | | |
|---|---|---|---|---|---|---|---|---|
| | avg | 95th | 98th | 99th | avg | 95th | 98th | 99th |
| Baseline | -2.77 | 0.33 | 2.27 | 4.75 | -1.89 | 1.47 | 3.52 | 6.15 |
| Contrastive | -2.82 | **-0.05** | **1.19** | **2.37** | -1.93 | **1.02** | **2.29** | **3.84** |
| PLM Re-Weighting | **-2.83** | 0.25 | 2.06 | 4.42 | **-1.94** | 1.38 | 3.32 | 5.72 |

(Makansi et al., 2021), and PLM re-weighted model (Kozerawski et al., 2022) for pedestrians are shown in Table 1. In this table, we also show the long-tailed most likely FDE metrics by reporting the 95th, 98th, and 99th percentile FDE error on the test set. These results confirm the findings of Makansi et al. (2021) and Kozerawski et al. (2022) by showing how much worse the performance on the long tail is compared to average performance. This gap is closed slightly by the Contrastive model, but the performance of the worst 1% of the data (i.e. the 99th percentile metric) is more than 5 times worse than average performance. Therefore, closing this gap by improving prediction of these long-tailed examples could greatly improve overall performance.

Furthermore, we compare the three methods using their KDE performance, as the KDE provides a way to compare multi-modal distributions of predictions. As can be seen from Table 2, both the contrastive method and the PLM re-weighting method improve pedestrian prediction on average, and in the long tail. Although the contrastive method focuses more on the long tail and improves average performance less as a result, the PLM re-weighting method focuses more on maintaining high average performance and consequently improves the long-tailed performance slightly less. These conclusions are also supported by the performance histograms in Figures 2a and 2b. While the PLM re-weighting method's evaluation shows more examples within the lowest error bin (KDE of less than -5) the contrastive method didn't yield any examples with errors that low. Meanwhile, the contrastive method yielded fewer examples with KDE errors greater than 1. Interestingly, the performance of the PLM re-weighting method is more 'long-tailed' than that of the baseline method, in that the highest KDE of any example in the PLM method is 16.10, while that of the baseline is 13.56. However, this one example may be an outlier since the PLM re-weighting method yields fewer examples with a KDE higher than 10 than the baseline.

These results confirm that the good results of Makansi et al. (2021) and Kozerawski et al. (2022) persist even when their methods are applied to Trajectron++ (Salzmann et al., 2021), a model which predicts a probability distribution of future trajectories with a likelihood associated with each prediction. However, contrary to the results in Kozerawski et al. (2022), the PLM re-weighting method does not outperform the contrastive method on long-tailed metrics when the KDE performance of pedestrians is taken into account.

Results for pedestrians in NuScenes shows that both long-tailed learning methods examined improve average performance as well as long-tailed performance, and that the contrastive method improves long-tailed performance more while the PLM re-weighting method improves average performance more.

Table 3: Vehicle FDE of most likely trajectories on NuScenes, predicting 3s and 4s into the future, where the columns are the average performance across the test set, and the 95th, 98th, and 99th percentile error across the test set

| Model | Vehicle FDE @3s | | | | Vehicle FDE @4s | | | |
|---|---|---|---|---|---|---|---|---|
| | avg | 95th | 98th | 99th | avg | 95th | 98th | 99th |
| Baseline | 1.14 | 3.99 | 5.25 | **6.28** | 2.21 | 7.69 | 9.97 | 11.68 |
| Contrastive | 1.18 | 4.17 | 5.47 | 6.43 | 2.25 | 7.95 | 10.34 | 12.06 |
| PLM Re-Weighting | **1.10** | **3.95** | **5.24** | **6.28** | **2.11** | **7.61** | **9.82** | **11.27** |

Table 4: Vehicle KDE on NuScenes, predicting 3s and 4s into the future, where columns are the average performance across the test set, and the 95th, 98th, and 99th percentile error across the test set

| Model | Vehicle KDE @3s | | | | Vehicle KDE @4s | | | |
|---|---|---|---|---|---|---|---|---|
| | avg | 95th | 98th | 99th | avg | 95th | 98th | 99th |
| Baseline | -1.61 | **2.02** | 3.54 | **5.18** | -0.71 | 3.12 | **4.68** | **6.86** |
| Contrastive | -1.64 | 2.08 | **3.43** | 5.19 | -0.74 | **3.11** | 4.75 | 6.97 |
| PLM Re-Weighting | **-1.71** | 2.55 | 4.65 | 6.56 | **-0.82** | 3.62 | 5.84 | 8.23 |

### 4.2.2 VEHICLES

Since the metrics reported by Makansi et al. (2021) and Kozerawski et al. (2022) use a non-intuitive, difficult to visualize definition of FDE, (minFDE), we re-report the FDE metric in terms of most likely FDE (the final distance error of the single most likely prediction). The most-likely FDE performances of the baseline model (Salzmann et al., 2021), contrastive model (Makansi et al., 2021), and PLM re-weighted model (Kozerawski et al., 2022) for vehicles are shown in Table 3. Similarly to the pedestrian FDE results, these results also confirm that the performance on the long tail is much worse than the average performance by a large factor. Although it seems like the PLM re-weighting method is the best of the three methods by most-likely FDE, the most-likely metric does not take into account the fact that the future is often multimodal: many futures may be equally probable while only one future plays out and gets recorded as ground truth (Mangalam et al., 2020). Therefore, the most-likely metric can easily be optimized by always predicting a 'mean' trajectory (Pajouheshgar & Lampert, 2018) that looks nothing like the trajectories in the various modes of the distribution (e.g. instead of predicting either a right-turn or a straight path, a model can predict an unlikely diagonal path and get better results on the most-likely FDE). Therefore, we only compare models on the KDE metric, which measures the accuracy of a distribution of trajectories instead of that of a single trajectory.

The KDE performances of vehicles in the NuScenes dataset (see Table 4), show that the PLM re-weighting method actually performs worse than the baseline on all the long-tail metrics. This analysis is supported by Figure 2d which shows that the PLM re-weighting method yields many more examples whose predicted trajectory distributions have a KDE of greater than 5. This, in combination with the model's good performance on the most-likely FDE long-tail metrics, shows that for long-tail examples, it is likely that the PLM re-weighting model is predicting a 'mean' long-tail trajectory instead of a multimodal distribution of more accurate trajectories.

Furthermore, it seems that for vehicles, neither model reliably out-performs the baseline on long-tailed KDE metrics. Both long-tail techniques improve average performance by improving non-long-tailed predictions, as shown by Figures 2c and 2d (which show more examples with KDEs of less than 5 than the baseline). In turn, they both also regress long-tailed performance in many cases. This shows that neither method is effective at improving long-tailed prediction for vehicles.

### 4.3 QUALITATIVE EVALUATION

In order to further investigate the differences in performance between the three models, we also perform a qualitative evaluation, as shown in Figure 3. Although this figure highlights instances where

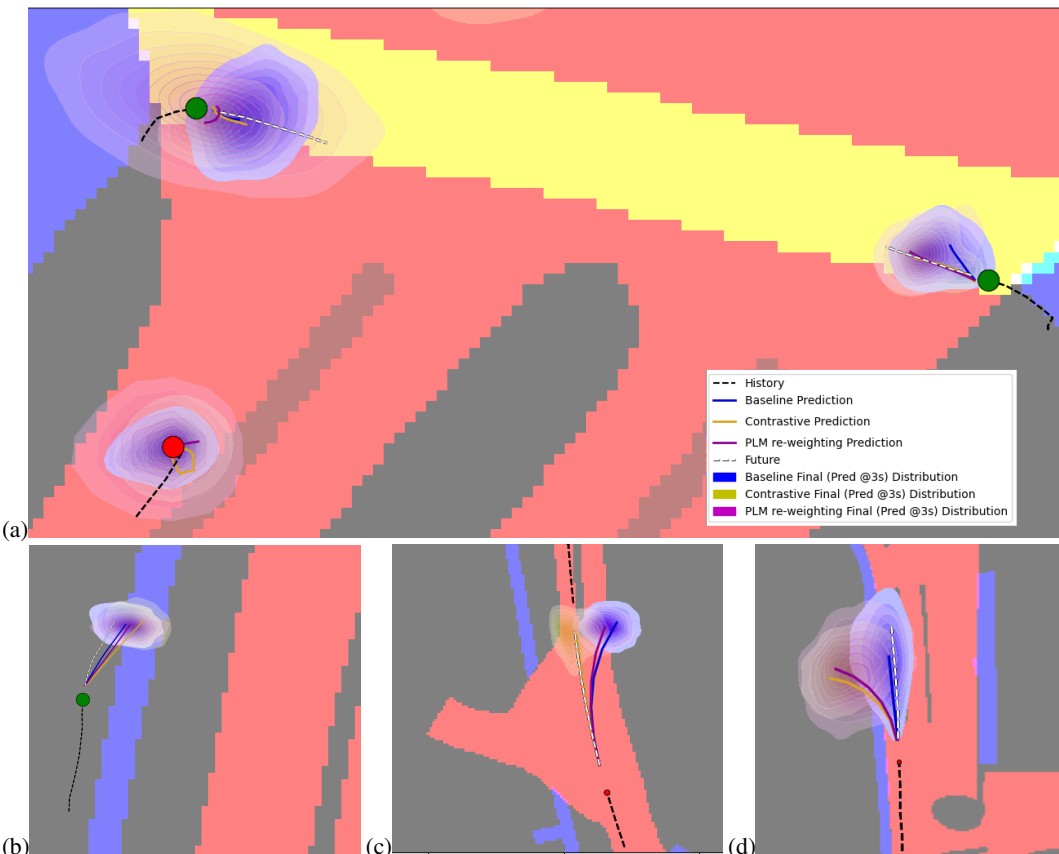

Figure 3: Qualitative evaluation for vehicles (red dots) and pedestrians (green dots). For each example shown, the most-likely predicted trajectory by each model is shown by the colored lines, while the probability distribution of the predicted position at a timestep of 3s into the future is shown by the filled contours, with each color representing a different model. (a) shows an example where contrastive and PLM re-weighting methods perform better than the baseline (for the top-right pedestrian in the image). (b) shows an example where the contrastive and PLM re-weighting methods perform worse than the baseline on a pedestrian. (c) shows an example where the contrastive method performs much better than both the PLM re-weighting and baseline methods on a vehicle. (d) shows an example where both the contrastive and PLM re-weighting methods perform worse than the baseline. Better performance is indicated by a prediction that is closer to the ground truth future (i.e. the dotted white line).

the baseline prediction and the Contrastive or PLM model predictions differ significantly, we observed that in the majority of cases, all models predicted paths that were fairly similar, showing that more work needs to be done to introduce models that predict different types of paths. Furthermore, the PLM re-weighting model typically predicted a path that lay in between the baseline model's prediction and the contrastive model's prediction, showing that it is the more moderate long-tailed learning method. Finally, in many cases (for example, in the case of the pedestrian in the top left of Figure 3a), the distribution of predictions for the contrastive loss model had a higher variance than that of the PLM re-weighting model and baseline models. This shows that the diversity of predictions made by the contrastive loss model is greater, leading to better long-tailed predictions in some cases.

## 5 CONCLUSION

In conclusion, we find that for pedestrians, the contrastive and PLM re-weighting methods make improvements over the baseline both on average, and in the long tail, with the contrastive model making more improvements in the long tail and the PLM re-weighted model making more improvements on average. However, this does not prove to be the case for vehicles: as can be seen in Table 4, neither model makes reliable improvements on the baseline. These results are slightly different to the results reported in Makansi et al. (2021) and Kozerawski et al. (2022) since these works apply their techniques to a prediction method which does not predict a distribution of trajectories with associated likelihoods for each prediction. Since our work re-evaluates these techniques on a prediction method which predicts a likelihood for each trajectory, we can compare these methods by the KDE metric, which measures the accuracy of the entire predicted distribution more reliably than the minADE/minFDE metric Pajouheshgar & Lampert (2018).

## 6 FUTURE WORK

Due to the methods' inability to improve the long-tail performance of vehicles as they have for pedestrians, more work needs to be done to understand the differences between pedestrian prediction and vehicle prediction, and improve long-tailed vehicle prediction accordingly. One major difference is that vehicle prediction makes more use of the semantic map input Khakzar et al. (2020). Therefore, one way to improve long-tailed vehicle prediction may be to recognize areas on the map which are more prone to have improperly predicted vehicles in them (i.e. vehicles within the low performing long tail of the dataset) and re-weight those areas accordingly, such that the model can better differentiate between easy and difficult examples based on their location.

Furthermore, the contrastive model only differentiates between 'easy' and 'difficult' examples, whereas there are many reasons why an example may be 'difficult'. This lack of differentiation can be seen in the PLM re-weighting model's vehicle performances, where the model seemed to be predicting the 'mean' long-tailed future instead of a multi-modal future corresponding to different reasons for difficulty. Splitting 'difficult' examples into separate categories may help the model better predict the different modes of behaviors within the long tail instead of simply predicting the 'mean' trajectory of long-tailed eamples.

Future work will include experiments to determine how re-weighting long-tailed locations on the map can help long-tailed vehicle performance, and how creating an ensemble learning model which can predict the different modes of the long tail can better learn to predict a distribution that accurately models each mode.

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

## A  APPENDIX

In addition to reproducing the models from Makansi et al. (2021) and Kozerawski et al. (2022), we also perform an ablation study to test different hyperparameters and the extents to which the regularization factor can be applied. These results can be seen in Table 5. As can be seen, the best contrastive loss model still uses the original regularization factor from Makansi et al. (2021) (1x), and the best PLM re-weighting model still uses the original regularization factor from Kozerawski et al. (2022) (100%).

Table 5: Ablation study of different factors of contrastive and PLM re-weighting loss used. After a hyper parameter search, we found best results using $\eta = 100$ and an $\xi = 0.01$ for the PLM re-weighting loss (see PLM loss formula in Figure 1). The percentages indicate how much of the PLM loss is added to the original loss: the percentage $p$ indicates that $loss_{total} = (1 - p) * loss_{baseline} + p * loss_{plm}$. For the contrastive method, Makansi et al. (2021) used a factor of 1 for the contrastive loss, where $loss_{total} = loss_{baseline} + loss_{contrastive}$ so we also train an additional model to have a $loss_{total}$ of $loss_{baseline} + 10 * loss_{contrastive}$, with a factor of 10 in order to observe its performance.

| Model | Pedestrian FDE @3s | | | | Pedestrian FDE @4s | | | | Pedestrian KDE @3s | | | | Pedestrian KDE @4s | | | |
|---|---|---|---|---|---|---|---|---|---|---|---|---|---|---|---|---|
| | avg | 95th | 98th | 99th | avg | 95th | 98th | 99th | avg | 95th | 98th | 99th | avg | 95th | 98th | 99th |
| PLM 25% | 0.37 | **1.12** | 1.57 | 1.91 | 0.61 | 1.90 | 2.59 | 3.08 | -2.61 | 0.42 | 2.33 | 4.73 | -1.76 | 1.59 | 3.81 | 6.14 |
| PLM 50% | 0.37 | **1.12** | 1.54 | **1.85** | 0.62 | 1.88 | **2.49** | **2.95** | -2.78 | 0.16 | 2.00 | 4.08 | -1.91 | 1.30 | 3.30 | 5.60 |
| PLM 75% | **0.36** | 1.13 | 1.54 | 1.86 | 0.60 | 1.88 | 2.56 | 3.01 | -2.81 | 0.28 | 2.22 | 4.61 | -1.94 | 1.41 | 3.33 | 6.09 |
| PLM 100% | **0.36** | 1.15 | 1.54 | 1.87 | **0.59** | 1.91 | 2.56 | 3.01 | **-2.83** | 0.25 | 2.06 | 4.42 | -1.94 | 1.38 | 3.32 | 5.72 |
| Contrastive 10x | 0.37 | 1.15 | 1.55 | 1.90 | 0.60 | 1.91 | 2.57 | 3.06 | **-2.83** | 0.08 | 1.58 | 3.18 | **-1.95** | 1.18 | 2.91 | 4.57 |
| Contrastive 1x | **0.36** | **1.12** | 1.53 | 1.86 | 0.60 | 1.87 | 2.55 | 3.01 | -2.82 | **-0.05** | **1.19** | **2.37** | -1.93 | **1.02** | **2.29** | **3.84** |

| Model | Vehicle FDE @3s | | | | Vehicle FDE @4s | | | | Vehicle KDE @3s | | | | Vehicle KDE @4s | | | |
|---|---|---|---|---|---|---|---|---|---|---|---|---|---|---|---|---|
| | avg | 95th | 98th | 99th | avg | 95th | 98th | 99th | avg | 95th | 98th | 99th | avg | 95th | 98th | 99th |
| PLM 25% | 1.11 | 3.93 | **5.19** | 6.32 | 2.12 | 7.55 | 9.76 | 11.36 | -1.69 | 2.22 | 3.95 | 5.68 | -0.79 | 3.31 | 5.18 | 7.74 |
| PLM 50% | 1.14 | 4.02 | 5.32 | 6.28 | 2.16 | 7.68 | 9.89 | 11.56 | -1.69 | 2.13 | 3.92 | 5.58 | -0.77 | 3.14 | 4.89 | 7.19 |
| PLM 75% | **1.08** | **3.91** | 5.21 | **6.16** | **2.07** | **7.34** | **9.59** | **11.30** | -1.59 | 2.62 | 4.61 | 6.56 | -0.67 | 3.63 | 5.74 | 8.13 |
| PLM 100% | 1.10 | 3.95 | 5.24 | 6.28 | 2.11 | 7.61 | 9.82 | **11.27** | **-1.71** | 2.55 | 4.65 | 6.56 | **-0.82** | 3.62 | 5.84 | 8.23 |
| Contrastive 10x | 1.16 | 4.18 | 5.61 | 6.76 | 2.18 | 7.82 | 10.30 | 12.28 | -1.44 | 2.32 | 3.62 | **4.89** | -0.55 | 3.30 | 4.85 | **6.63** |
| Contrastive 1x | 1.18 | 4.17 | 5.47 | 6.43 | 2.25 | 7.95 | 10.34 | 12.06 | -1.64 | **2.08** | **3.43** | 5.19 | -0.74 | **3.11** | **4.75** | 6.97 |

