# OpenReview forum: "In-Depth Comparison of Regularization Methods For Long-Tailed Learning in Trajectory Prediction"
_ICLR.cc/2024/Conference — Submitted to ICLR 2024_

### Official Review · Reviewer_bsJi · 2023-10-25

**Soundness:** 2 fair
**Presentation:** 2 fair
**Contribution:** 2 fair
**Rating:** 3
**Confidence:** 4

**Summary:**

This paper targets the long-tail trajectory prediction problem, and it compares two regularization methods for long-tail learning, contractive loss (Makansi et al.) and PLM loss (Kozerawski et al.). The authors implemented the two regularization methods on the Trajectron++ model and evaluated them against the baseline on the nuScenes dataset.

The authors showed both quantitative results and qualitative examples of the two long-tail learning methods. The results show that the contrastive and PLM re-weighting methods make improvements for both average and long-tail cases for pedestrian prediction. However, they do not improve vehicle prediction.

**Strengths:**

* The authors provided comprehensive evaluation results on a public benchmark.

**Weaknesses:**

* This paper does not have sufficient research innovation to be accepted at ICLR. The authors simply took two existing long-tail learning methods and implemented them on an existing trajectory prediction model.

* The evaluation result is weak. The performance of the proposed model has little improvement over the baseline.

**Questions:**

N/A

---

### Official Review · Reviewer_yYGK · 2023-10-28

**Soundness:** 2 fair
**Presentation:** 2 fair
**Contribution:** 1 poor
**Rating:** 3
**Confidence:** 4

**Summary:**

This paper deals with the problem of long-tailed learning in trajectory prediction of traffic agents, and compares two regularization methods.
The first method uses contrastive loss to distinguish the features of uncommon traffic behaviors from common ones, and the second method uses PLM re-weighting loss to assign more weight to rare examples with large errors.
The paper measures the performance of these methods on the NuScenes dataset for pedestrians and vehicles.

**Strengths:**

The paper discusses the effects of the regularization methods, and visualizes the experimental results.

**Weaknesses:**

The writing should be improved a lot.
* In Section 2.3, line 6, it appears to reference a figure from another paper. This is unusual and may not be accepted for any venue.
* In Sections 3.2.1, 3.2.2, and other parts, there is a lack of equations or explanations about the loss functions, making it difficult to understand how they are computed.
* The latest trajectory prediction papers (2023~) are not referenced enough. It will be better to add them to the reference part.

The contribution is not significant. Applying and measuring existing methods to existing metrics on existing datasets is not enough for ICLR standards. The analysis is not extensive or insightful, and it does not propose any concrete solutions for improving long-tailed prediction.

**Questions:**

* In Section 3.2.2, line 3, what is the kurtosis term? There is no mention of it before, and it is unclear what it means.

* In Section 4.1, last third line, why is it important to evaluate all possible future distributions? For long-tailed problems, the whole distribution may not follow the ground truth. It may be more appropriate to use FDE or ADE metrics that evaluate whether one of K predictions captures the long tail.

---

### Official Review · Reviewer_wvR7 · 2023-10-31

**Soundness:** 2 fair
**Presentation:** 2 fair
**Contribution:** 2 fair
**Rating:** 3
**Confidence:** 5

**Summary:**

The paper focuses on evaluating long-tailed learning techniques in the context of pedestrian trajectory prediction. The authors investigate two specific methods: Contrastive Loss and PLM Loss and apply them to the Trajectron++ model. The evaluation is conducted using the NuScenes dataset, considering both average performance and performance in the long tail of the distribution. The results indicate that both methods improve average performance and long-tailed performance for pedestrian trajectory prediction, with the Contrastive Loss method showing more significant improvements in the long tail.

**Strengths:**

- Comprehensive Evaluation: The paper provides a thorough evaluation of long-tailed learning techniques, considering both average performance and performance in the long tail of the distribution.

- Clear Methodology: The authors clearly describe the methods being evaluated and the modifications made to apply them to the Trajectron++ model.

**Weaknesses:**

- Limited to Pedestrian Trajectory Prediction: The evaluation is limited to pedestrian trajectory prediction, and it is not clear how well the findings would generalize to other types of trajectory prediction.

- Dependence on Specific Models: The evaluation is conducted using specific models (Trajectron++), and the results might not be applicable to other trajectory prediction models. More recent baselines should be used to evaluate the proposed method.

- Potential for More In-Depth Analysis: While the paper provides a good overview of the methods and their performance, there could be a more in-depth analysis of why certain methods perform better in specific scenarios.

- Unclear Presentation: Figure 1 and Figure 3 are not clear, some screenshots are used in Figure 1 to indicate the equations, which is not appropriate in submission.

**Questions:**

- How well do you expect the findings of this paper to generalize to other trajectory prediction models?

- Have you considered evaluating the methods on other datasets, and if so, do you expect the results to be consistent with the findings from the NuScenes dataset?

- What insights can you provide on the characteristics of the long-tailed distribution of trajectory prediction errors, and how do these characteristics influence the choice of long-tailed learning techniques?

- How do you balance the need for high average performance with the need for improved performance in the long tail of the distribution?

---

### Official Review · Reviewer_2aGX · 2023-11-03

**Soundness:** 3 good
**Presentation:** 3 good
**Contribution:** 2 fair
**Rating:** 5
**Confidence:** 4

**Summary:**

The authors compare how regularization can be used to improve long-tailed motion prediction. The two regularizers that the authors study are a contrastive learning approach and a data re-weighting approach. The authors report metrics for these methods applied to the trajectron model trained on the nuScenes dataset for vehicles and pedestrians. The authors find that while performance improves along a variety of metrics for pedestrians, the improvement for vehicles is less significant, pointing to the need for future work that can improve performance across the board.

**Strengths:**

I found the paper very easy to read and the contributions clearly stated. The metrics that the authors choose to use for analysis are principled - e.g. the final distance error and estimated kernel density across different percentile thresholds. The numbers themselves seem reasonable, and Figure 2 is very helpful for understanding the effect of the regularizers across classes as a complement to Tables 1-4. Figure 1 makes it clear where the authors choose to add the contrastive loss in the latent space, and where the re-weighting occurs at the model output.

**Weaknesses:**

My only criticism of the paper is that I found it hard to extract actionable insights from the analysis that was presented. I think there are a few factors that currently limit the applicability of the results.

First, as an "in-depth study", I would hope for analysis of more methods than the two presented in the paper and on more datasets than nuScenes, which is a small dataset for the purposes of motion prediction. For instance, "Taming the Long Tail of Deep Probabilistic Forecasting" benchmarked 7 methods across 3 datasets, I was expecting the number of experiments for this paper to be in that range. Additionally, the current paper does not provide a sense of how the performance boost due to these methods scales with more data, which is a critical point given that most people in the machine learning community would expect regularizers to decrease in effectiveness with more data. I would highly recommend adding experiments on the waymo open dataset to benchmark scalability, as it has 100s of hours of data. I also think testing on a larger dataset will significantly decrease the variance of the CVaR metrics, since the lowest percentile of nuScenes trajectories for pedestrians contains a very small number of trajectories. If testing on a larger dataset is too difficult, error bars for the CVaR metrics would be very helpful to see.

Second, the impact of the paper would be improved significantly if the authors can dig more deeply into why the improvement for vehicles over the baseline is so much smaller than pedestrians. Is it due to the fact that there is more vehicle data in the training set, so the benefit of adding a regularizer is less? Is it that the benefit is primarily on low-velocity trajectories, and a similar effect appears if we look at low-velocity vehicle trajectories? Some more analysis would be helpful, especially since long-tailed vehicle prediction is so important for self-driving.

A few other edits:
- use \citep{} instead of \cite{} unless referring directly to the paper
- Section 3.2 - multi-model -> multi-modal

**Questions:**

My main questions, as mentioned in the "weaknesses" section, are related to figuring out actionable insights that can be gained from the analysis in the paper. I think the experiments are solid and the metrics are principled, but I don't yet see what the takeaway is that can improve the performance of motion prediction models on the long-tail. If the authors add more extensive evaluation of methods that improve long-tail performance, or offer insight into how these regularizers improve performance as a function of dataset size, I think it would greatly boost the impact of the paper.

---

### Meta-Review · Area_Chair_NRUL · 2023-12-15

**Metareview:**

The authors study the use of contrastive learning and data re-weighting approaches in improving long-tailed motion prediction. They applied these methods to the Trajectron++ model on the nuScenes dataset for vehicles and pedestrians. Results showed that while pedestrian performance improved, vehicle performance was less significant. Contrastive Loss and PLM Loss were also evaluated for pedestrian trajectory prediction. Contrastive Loss showed more significant improvements in the long tail. The authors also compared two regularization methods for long-tail trajectory prediction: contractive loss and PLM loss. Results showed that contrastive and PLM re-weighting methods improved pedestrian prediction, but not vehicle prediction.

## Strenghts

The paper provides a comprehensive evaluation of long-tailed learning techniques, considering both average and long tail performance. The authors use principled metrics like final distance error and kernel density, and provide clear methodology for applying these methods to the Trajectron++ model. The numbers used are reasonable and the paper's clear presentation is helpful.

## Weaknesses


• The study's applicability is limited by the lack of extensive analysis and datasets.
• The paper's focus on pedestrian trajectory prediction is limited, making it unclear how findings would generalize to other types of trajectory prediction.
• The study relies on specific models, which may not be applicable to other trajectory prediction models.
• The paper could benefit from more in-depth analysis of why certain methods perform better in specific scenarios.
• The presentation of the methods is unclear, with some screenshots used to indicate equations.
• The paper references a figure from another paper, which may not be accepted for any venue.
• The lack of equations or explanations about loss functions makes it difficult to understand how they are computed.
• The latest trajectory prediction papers (2023~) are not referenced enough.
• The contribution is not significant, as applying and measuring existing methods to existing metrics on existing datasets is not enough for ICLR standards.

**Justification For Why Not Higher Score:**

Contributions were not enough although results are acceptable.

**Justification For Why Not Lower Score:**

N/A

---

### Decision · Program_Chairs · 2024-01-16

Reject